# Prolonged QT Interval in SARS-CoV-2 Infection: Prevalence and Prognosis

**DOI:** 10.3390/jcm9092712

**Published:** 2020-08-21

**Authors:** Núria Farré, Diana Mojón, Marc Llagostera, Laia C. Belarte-Tornero, Alicia Calvo-Fernández, Ermengol Vallés, Alejandro Negrete, Marcos García-Guimaraes, Yolanda Bartolomé, Camino Fernández, Ana B. García-Duran, Jaume Marrugat, Beatriz Vaquerizo

**Affiliations:** 1Department of Cardiology, Hospital del Mar, 08003 Barcelona, Spain; 62834@parcdesalutmar.cat (D.M.); 62205@parcdesalutmar.cat (M.L.); 62547@parcdesalutmar.cat (L.C.B.-T.); 61725@parcdesalutmar.cat (A.C.-F.); 99056@parcdesalutmar.cat (E.V.); negrete_alejandro@hotmail.com (A.N.); marcos.garcia.guimaraes@gmail.com (M.G.-G.); 97096@parcdesalutmar.cat (Y.B.); 60851@parcdesalutmar.cat (C.F.); abgarciaduran@psmar.cat (A.B.G.-D.); 97912@parcdesalutmar.cat (B.V.); 2Department of Medicine, School of Medicine, Univ Autonoma de Barcelona, 08003 Barcelona, Spain; 3Heart Diseases Biomedical Research Group (GREC), IMIM (Hospital del Mar Medical Research Institute), 08003 Barcelona, Spain; 4REGICOR (Registre Gironí del Cor) Study Group, IMIM (Hospital del Mar Medical Research Institute), 08003 Barcelona, Spain; jaume@imim.es; 5CIBER (Centro de Investigación Biomédica en Red) of Cardiovascular Diseases (CIBERCV), Instituto de Salud Carlos III (ISCIII), 08003 Barcelona, Spain

**Keywords:** QT interval, COVID-19, hydroxychloroquine, azithromycin, prognosis, death

## Abstract

Background: The prognostic value of a prolonged QT interval in SARS-Cov2 infection is not well known. Objective: To determine whether the presence of a prolonged QT on admission is an independent factor for mortality in SARS-Cov2 hospitalized patients. Methods: Single-center cohort of 623 consecutive patients with positive polymerase-chain-reaction test (PCR) to SARS Cov2, recruited from 27 February to 7 April 2020. An electrocardiogram was taken on these patients within the first 48 h after diagnosis and before the administration of any medication with a known effect on QT interval. A prolonged QT interval was defined as a corrected QT (QTc) interval >480 milliseconds. Patients were followed up with until 10 May 2020. Results: Sixty-one patients (9.8%) had prolonged QTc and only 3.2% had a baseline QTc > 500 milliseconds. Patients with prolonged QTc were older, had more comorbidities, and higher levels of immune-inflammatory markers. There were no episodes of ventricular tachycardia or ventricular fibrillation during hospitalization. All-cause death was higher in patients with prolonged QTc (41.0% vs. 8.7%, *p* < 0.001, multivariable HR 2.68 (1.58–4.55), *p* < 0.001). Conclusions: Almost 10% of patients with COVID-19 infection have a prolonged QTc interval on admission. A prolonged QTc was independently associated with a higher mortality even after adjustment for age, comorbidities, and treatment with hydroxychloroquine and azithromycin. An electrocardiogram should be included on admission to identify high-risk SARS-CoV-2 patients.

## 1. Introduction

Previous reports have highlighted the potential risk of cardiac complications and arrhythmias in patients with severe acute respiratory syndrome coronavirus 2 (SARS-Cov-2) infection [1]. The presence of a prolonged QT interval can further worsen prognosis. However, most of the information about the prognostic role of QT interval in SARS-Cov-2 infection has been derived from studies analyzing the effects of the treatment with hydroxychloroquine and azithromycin [2,3,4,5,6], a treatment associated with QT interval prolongation. The benefits of these treatments on prognosis are currently controversial.

Baseline QT interval abnormalities in the setting of SARS-Cov-2 infection can be secondary to the viral infection per se, the inflammatory state associated with SARS-Cov-2 infection, and ischemia or hypoxia [1]. Indeed, several viral infections like human immunodeficiency virus (HIV) and dengue have been independently associated with a prolonged QT interval [7,8,9]. Interestingly, acute coronavirus infection has been associated with a prolonged QT interval in rabbits [10], which suggests that the virus might have a direct effect on the heart. On the other hand, in the absence of infection, systemic inflammation and elevated C-reactive protein (CRP) have also been associated with QT prolongation [11,12,13,14,15,16]. These associations seem to be mediated, at least in part, by elevated interleukin-6 (IL-6) levels. Treatment with tocilizumab, an anti-IL-6 receptor antibody, has been associated with QT interval shortening [12,13]. Intriguingly, in men with HIV infections, those with elevated IL-6 had more prolonged QT [17], suggesting a potential additive effect of infection and inflammation on the QT interval. Thus, the presence of a prolonged QT interval on admission might be a marker of worse prognosis irrespective of the treatment the patients receive. Therefore, the aim of this study was to test the hypothesis that the presence of prolonged QT on admission is an independent factor for mortality in patients with SARS-Cov-2 infection.

## 2. Experimental Section

A single-center cohort study conducted at *Hospital del Mar*, Barcelona, Spain, from 27 February to 7 April 2020. Patients were followed up until 10 May 2020. All consecutive patients with laboratory-confirmed COVID-19 by means of polymerase-chain-reaction (PCR) test were included in the study. We collected demographic data, laboratory findings, comorbidities, and treatment received.

Baseline electrocardiogram (ECG) was defined as the ECG taken within the first 48 h after laboratory-confirmed COVID-19 diagnosis and always before the administration of any medication with a known effect on the QT interval. QT was automatically calculated as the time from the start of the Q wave to the end of the T wave and corrected for heart rate by the Bazett formula (QTc). All ECGs were done with the Philips PageWriter TC30 Cardiograph (Koninklijke Philips, Eindhoven, The Netherlands). Prolonged QTc was defined as a QTc > 480 milliseconds (ms) [18]. Although ECG was recommended in all patients, and especially in those who would receive medication that potentially modifies the QT interval, the decision to order the ECG was left to clinicians and adapted to the logistic capabilities of the center during the pandemic. Therefore, in the current analysis, we focused on patients who had a baseline ECG (Figure 1). However, patients who had a baseline ECG were also compared to those who did not have a baseline ECG. When patients had more than one ECG during hospitalization, maximum QTc interval was also collected. QTc prolongation was defined as an increase of at least one millisecond in QTc compared to baseline QTc.

According to the protocol at our center at the time of the study, treatment with hydroxychloroquine and azithromycin was recommended to all patients. Azithromycin was given once a day (500 mg) for three days and hydroxychloroquine was given five days at a dose of 400 mg twice a day the first day and 200 mg twice a day the following four days. This treatment was contraindicated when QTc was longer than 550 ms. If QTc was longer than 500 ms, a daily ECG was mandatory. The use of tocilizumab was decided based on the presence of pulmonary infiltrates on chest X-ray or worsening of previous infiltrates, PaO2/FiO2 <300, and at least one of these parameters: IL6 ≥ 40 ng/L (or PCR ≥ 100 mg/L), D dimer ≥ 1000 ng/mL, or ferritin ≥ 700 ng/mL.

The primary endpoint was all-cause death at 30 days after COVID-19 diagnosis.

This study was performed in accordance with the provisions of the Declaration of Helsinki, ISO 14155 and clinical practice guidelines. The study protocol was approved by the Institutional Ethics Committee and the hospital’s research commission (number CEIm 2020/9178). Oral informed consent was obtained, but the need for written informed consent was waived in light of the infectious disease hazard.

### Statistical Analysis

Categorical variables were summarized as number and percentages, and continuous variables were summarized as the mean and standard deviation (SD), or the median and interquartile range (IQR), depending on the variable distribution. Patients’ characteristics were compared between prolonged QTc (cut-off point > 480 ms) and outcome status categories (death) by Student’s *t*-test or Mann–Whitney U test for continuous variables, and by Pearson’s chi-squared test for categorical variables.

Kaplan–Meier survival estimates were used to calculate the 30-day observed cumulative incidence of death, and statistical significance was tested by the log-rank test. The adjusted hazard ratio (HR) of death for QTc status was analyzed using Cox proportional hazard models. The models were adjusted for potential confounders selected by stepwise forward inclusion, among patient characteristics that were significantly associated with a prolonged QTc status as well as with the composite endpoint (death). Because the number of end-points was low, it was not possible to include all variables with *p* < 0.05. We chose the variables with *p*-value < 0.001 and prevalence >5%, therefore moderate to severe valve heart disease was not included in the model (overall prevalence 3.7%). The variables included in the model were age, baseline QTc > 480 ms, chronic kidney disease, treatment with azithromycin and hydroxychloroquine, ischemic chronic disease, atrial fibrillation or flutter, heart failure, and the presence of any cardiovascular risk factor. We acknowledge that there might be a survival bias associated with treatment (or an immortal time bias) wherein you must survive long enough to be treated. However, since the treatment with HCQ and AZM are known to prolong QT and might predispose to ventricular arrhythmias, we thought that the inclusion of treatment in the model was warranted. However, in order to minimize the bias, we created a model with the same variables except did not include the treatment received. Second, we also did a sensitivity analysis excluding patients who died during the first 48 h of admission. Finally, standardized differences were calculated, and a difference >0.10 was considered clinically significant. In addition, *p*-values < 0.05 were considered statistically significant. All tests were performed with SPSS version 25 (IBM SPSS versión 25, Armonk, NY, USA).

## 3. Results

Sixty-one patients (9.8%) had prolonged QTc on admission. Only 20 patients (3.2%) had a baseline QTc > 500 ms. Baseline characteristics are described in Table 1. Briefly, patients with prolonged QTc were older and had more comorbidities. Moreover, they had higher levels of C-reactive protein, leucocytes, lactate, and procalcitonin. Similar results were seen in patients who died (Table 2).

Only 245 patients (39% of the cohort) had a follow-up ECG during hospitalization. Of those, 77 patients (31.4%) had the longest QTc interval on admission, whilst 68.6% had QTc prolongation during hospitalization. Baseline characteristics, treatment, and prognosis of patients who had QTc prolongation on follow-up ECG during hospitalization are described in Table 3. Interestingly, both baseline QTc duration (441.75 ± 38.5 ms vs. 435.38 ± 31.6 ms, *p* = 0.17) and the percentage of patients with baseline QTc > 480 ms (10.4% vs. 11.9%, *p* = 0.73) were similar in those who prolonged QTc during hospitalization compared with those who did not. As expected, patients with QTc interval prolongation during hospitalization had higher prescription of hydroxychloroquine and azithromycin.

In-hospital treatment and prognosis are shown in Table 4 and Table 5. There were no episodes of ventricular tachycardia or ventricular fibrillation during hospitalization. When analyzed by sex, the presence of QTc ≥ 480 ms was associated with a higher mortality in both sexes. In women, mortality was 56.7% (17/30) in those with QTc ≥ 480 ms compared with 8.4% (20/237) in the non-prolonged QTc interval, *p* < 0.001. Similar results were seen in men: mortality was 25.8% (8/31) in the prolonged QTc interval group vs. 8.9% (29/325) in the non-prolonged QTc interval group, *p* = 0.003. This cut-off was independently associated with death in women (univariable HR 8.53 (95% CI: 4.45–16.36), *p* < 0.001, multivariable HR 4.04 (1.98–8.27), *p* < 0.001), whereas there was a strong tendency in the same direction in men (univariate HR 2.27 (95% CI: 0.99–5.23), *p* = 0.053).

Mortality rate was much higher in patients with prolonged QTc at admission (41.0% vs. 8.7%, *p* < 0.001), as shown in Table 6 and the Kaplan–Meier survival curves in Figure 2. A baseline QTc > 480 ms was independently associated with higher mortality (HR 2.68 (1.58–4.55), *p* < 0.001). This result was similar when treatment was not included in the model (HR 2.78 (95% CI 1.66–4.66), *p* < 0.001). In a sensitivity analysis excluding patients who died during the first 48 h of admission (18 patients, 24.3% of all patients who died), the results were also similar (HR: 2.073 (95% CI: 1.073–4.005), *p* = 0.03).

The baseline characteristics, prognosis, and presentation of patients without a baseline ECG are summarized in Table 7. This group of patients was younger and had less cardiovascular risk factors and comorbidities. The clinical presentation was less severe and 24% were not treated with hydroxychloroquine or azithromycin. Death rate was similar to those with a baseline ECG (10.8 vs. 11.9, *p* = 0.66).

## 4. Discussion

In this study, we found that a prolonged QTc interval at admission is present in almost 10% of patients with SARS-CoV-2 infection. Even though these patients had more comorbidities and worse clinical profile at presentation, the presence of a prolonged QTc was independently associated with increased mortality.

The mean baseline QTc interval in our study was 437.0 ± 34.5 ms. Several studies in SARS-CoV-2 infection have reported similar baseline QTc intervals, with mean values ranging from 415 to 455 ms [2,3,4,5,6]. There are different definitions of prolonged QTc and the use of any of them might have affected the results of our study. We chose the cut-off value of 480 ms following the ESC Guidelines [18]. Although the prevalence of QTc > 480 ms was 9.8%, the prevalence of very prolonged QTc (QTc > 500 ms) was very low, only affecting 3.2% of patients and similar to other studies [6]. Therefore, using this restrictive cut-off as a screening tool would have had limited clinical value.

The majority of studies have focused on QTc interval and risk of arrhythmias, especially in the setting of hydroxychloroquine treatment. This treatment (with or without azithromycin) is associated with a prolongation of the QT interval in 2.8 to 18.9% of patients [2,4,6,19,20]. However, these results depend on the definition of QT prolongation used and the dose of hydroxychloroquine. Interestingly, the risk of ventricular arrhythmias was very low and, consistent with our results, several studies did not show any episode of *torsade de pointes* or arrhythmic death [2,4,6,20,21]. In rheumatologic disease studies, the use of hydroxychloroquine has also been associated with QTc interval prolongation but not to increased mortality [22]. Thus, if randomized controlled trials were to show increased survival in SARS-CoV-2 infection with this treatment, data available show that the fear of malignant arrhythmias should not be a deterrent to its use with proper QT interval monitoring.

However, the interest in the prognostic value of QTc interval goes beyond its potential interaction with treatment. The electrocardiogram (ECG) is a cheap non-invasive tool that can be found in all healthcare settings, from local clinics to tertiary hospitals. However, ECG is an underused tool in risk stratification. In our cohort, all-cause death was higher in patients with prolonged QTc (41.0% vs. 8.7%, *p* < 0.001, multivariable HR 2.68 (1.58–4.55), *p* < 0.001). As expected from previous research, age and comorbidities were associated with prolonged QTc interval and worse prognosis [23,24,25,26]. Some studies have shown that almost 20% of patients with chronic kidney disease (CKD) have a prolonged QTc interval than patients without CKD, and the presence of a prolonged QTc interval in this group is associated with increased cardiovascular and all-cause mortality [24]. Moreover, age *per se* is associated with a prolonged QT interval [23,27]. In patients with acute heart failure, the QTc interval has been associated with 5-year all-cause mortality in J-shape with nadir of 440 to 450 ms in male and 470 to 480 ms in female, although its significance decreased in females [28]. Similar results are seen in chronic heart failure, where the presence of prolonged QTc is also associated with higher mortality (41% vs. 14%, *p* = 0.001) [29]. In patients with prior cardiovascular disease, both cardiovascular mortality and sudden death were higher in patients with prolonged QTc, with relative risks that ranged from 1.1 to 3.8 for total mortality, from 1.2 to 8.0 for cardiovascular mortality, and from 1.0 to 2.1 for sudden death [30]. There are well documented sex-dependent differences in normal QT interval and age- and sex- specific cut-offs for prolonged QTc (>450 ms for men and >470 ms for women) have been proposed [27]. Therefore, by using a cut-off of 480 ms, it is possible that high-risk men were not identified. When analyzed separately by sex, we saw that patients with prolonged QTc had higher mortality (56.7% vs. 8.4% in women, *p* < 0.001, and 25.8% vs. 8.9% in men, *p* = 0.003). The 480 ms cut-off was independently associated with death in women, whereas there was a strong tendency in the same direction in men (univariate HR 2.27 (95% CI: 0.99–5.23), *p* = 0.053). It is worth mentioning that the number of events was very low in men (only eight patients in the QTc ≥ 480 ms died) and that might explain the lack of statistical significance in men. Although the use of a different cut-off according to sex could be useful, using several cut-off points depending on sex might not be feasible in clinical practice when different types of healthcare professionals at several levels of complexity are involved. The fact that patients with prolonged QTc had higher immune-inflammatory parameters and cardiac biomarkers (i.e., C-reactive protein, white blood cell count, serum lactate, procalcitonin, lactate dehydrogenase, D-dimer, troponin T, and NTproBNP) is intriguing. Although these differences could be due to a more severe presentation in a group of elderly comorbid patients, SARS-CoV-2 infection could be the cause of this prolonged QTc interval, either as a direct effect of the virus or through systemic inflammation. Studies done in rabbits showed that coronavirus infection was associated with QT interval prolongation [10], and coronavirus infection caused right and left ventricular dilation, myocardial fibrosis, and myocarditis [31,32]. Similarly to what had been observed in the animal model, echocardiograms done in patients with SARS-CoV-2 infection have shown a predominant right ventricular dilation, which was associated with increased troponin levels and worse prognosis [33]. On the other hand, several studies have described abnormal immune-inflammatory response to SARS-CoV-2 infection. A recent study has shown that levels of interleukin (IL)-1β, IL-6, IL-8, IL-10 and soluble TNF receptor 1 (sTNFR1) were all increased in patients with SARS-CoV-2 infection compared to healthy volunteers and cytokine ratios may predict outcomes in this population [34]. A recent meta-analysis has shown that other immune-inflammatory parameters, such as C-reactive protein, white blood cell count, and procalcitonin, were higher in severe SARS-CoV-2 infection compared to milder presentations [35]. Given that inflammation can also lead to QT interval prolongation [14,15], it is possible that SARS-CoV-2 infection prolongs a QTc interval through an inflammatory response. Hence, a prolonged QTc interval in SARS-CoV-2 infection could be the result of direct virus activity or be mediated by inflammation, which would help explain why a prolonged QTc is independently associated with 30-day mortality.

### Limitations

There are some limitations to our study. First, asymptomatic patients were not included in this registry, which confers a selection bias. Second, although this is the largest study assessing QTc prognostic value in SARS-CoV-2 infection, this is a single-center study with a limited number of patients. Third, data on prolongation of QTc during hospitalization should be viewed with caution because only 39% of patients had a repeated ECG during hospitalization, hence the risk of bias is potentially high. Fourth, the measurement of the QT interval can be difficult [27,36]. Previous studies have shown that only 60% of physicians were able to accurately measure a sample QT interval, even though the majority stated that their area of specialization was cardiology [37]. Several studies have shown that automated QTc measurements are accurate in comparison with manual QTc measurements [6,22,38]. Therefore, the use of automated ECG measurement is likely to offer greater accuracy and allow a wider use of this tool in all healthcare levels than the manual assessment. Finally, we cannot exclude that some of the deaths might be due to ventricular tachycardia or ventricular fibrillation that went unnoticed and were ultimately attributed to other causes.

## 5. Conclusions

Up to 10% of patients with SARS-CoV-2 infection had a prolonged QTc interval (i.e., >480 milliseconds) on admission. A prolonged QTc was independently associated with a higher risk of mortality even after adjustment for age, comorbidities, and treatment with hydroxychloroquine and azithromycin. Thus, the QTc interval should be measured in all patients with SARS-CoV-2 infection as a non-invasive and low cost tool for identifying high-risk patients.

## Figures and Tables

**Figure 1 jcm-09-02712-f001:**
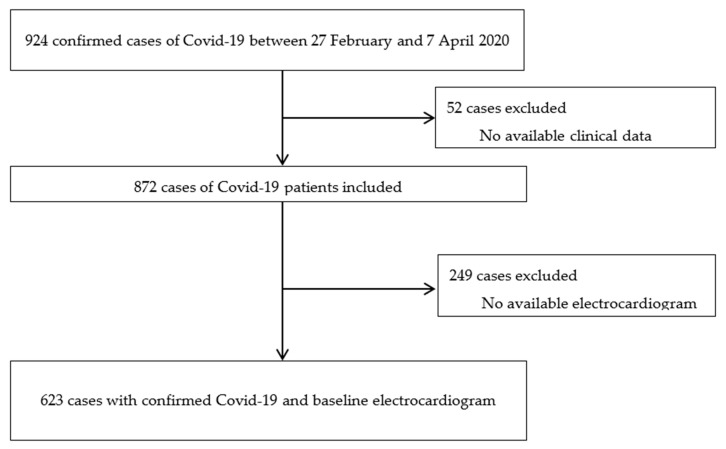
Flowchart of patient selection.

**Figure 2 jcm-09-02712-f002:**
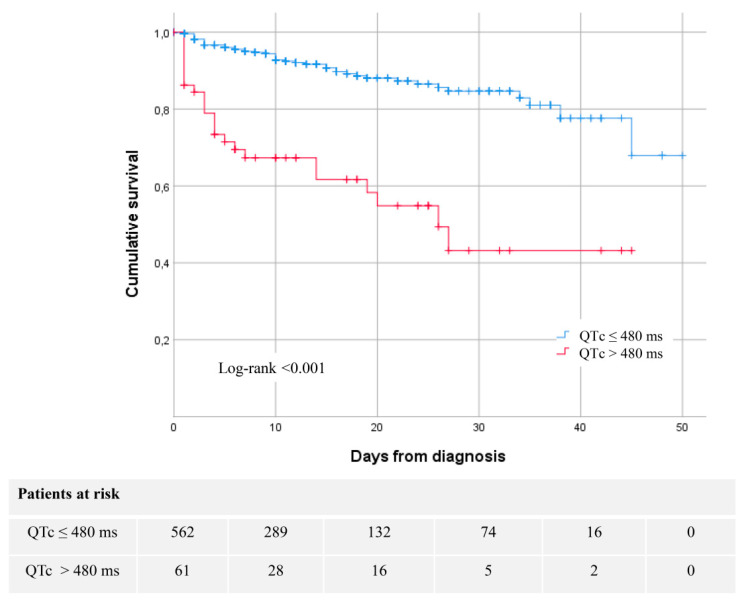
Kaplan–Meier 30-day survival curves for mortality by QTc during the time from medical contact.

**Table 1 jcm-09-02712-t001:** Comparison of baseline characteristics and clinical presentation between patients with and without prolonged QTc on admission.

	QTc ≤ 480 (*n* = 562)	QTc > 480 (*n* = 61)	*p*-Value	Standardized Differences
Women	237 (42.2)	30 (49.2)	0.29	0.14107
Age, years	62.9 ± 16.9	76.6 ± 12.6	<0.001	−0.92172
Diabetes	88 (15.7)	30 (49.2)	<0.001	0.76703
Hypertension	244 (43.4)	46 (75.4)	<0.001	0.68911
Dyslipidemia	179 (31.9)	33 (54.1)	<0.001	0.46121
CV Risk factors	360 (64.1)	55 (90.2)	<0.001	0.65377
Obesity	97 (21.7)	15 (28.8)	0.239	0.16617
Ischemic chronic disease	36 (6.4)	15 (24.6)	<0.001	0.51915
Atrial fibrillation or flutter	37 (6.6)	15 (24.6)	<0.001	0.51245
Heart failure	20 (3.6)	13 (21.3)	<0.001	0.55858
Moderate to severe valve heart disease	14 (2.5)	9 (14.8)	<0.001	0.44769
COPD	42 (7.5)	10 (16.4)	0.017	0.27780
Cancer	64 (11.4)	14 (23.0)	0.010	0.31028
CKD	39 (6.9)	18 (29.5)	<0.001	0.61132
Peripheral vascular disease	20 (3.6)	7 (1.5)	0.004	0.30370
Stroke	27 (4.8)	8 (13.1)	0.007	0.29411
Systolic blood pressure, mmHg	128.3 ± 19.0	130.4 ± 24.1	0.43	−0.09693
Diastolic blood pressure, mmHg	77.1 ± 13.3	74.5 ± 16.4	0.23	0.17954
Heart rate, bpm	91.2 ± 18.0	88.3 ± 21.5	0.27	0.13995
Respiratory rate, rpm	24.5 ± 7.1	25.6 ± 7.1	0.37	−0.15164
Oxygen saturation, %	95.1 ± 4.4	93.8 ± 6.1	0.13	0.23793
Baseline FiO2, %	25.9 ± 16.0	33.3 ± 25.4	0.033	−0.34956
PaO2/FiO2 < 300, n (%)	180 (42.8)	30 (60.0)	0.020	0.35028
Hemoglobin, g/dL	13.6 ± 1.6	12.5 ± 2.0	<0.001	0.59990
White blood cell count, per µL	7.04 ± 3.54	8.99 ± 4.76	0.003	−0.46572
Lymphocytes, per µL	1.40 ± 2.2	1.74 ± 2.64	0.35	−0.13720
Platelet count	209.0 ± 78.3	223.1 ± 109.0	0.33	−0.14919
Creatinine, mg/dL	1.02 ± 0.56	1.66 ± 1.44	0.001	−0.58761
eGFR, mL/min/1.73 m^2^	83.0 ± 29.2	59.1 ± 32.0	<0.001	0.77871
Creatine phosphokinase, U/L	93 (57–170)	95 (47–290)	0.97	−0.18600
Serum lactate, mmoL/L	1.44 ± 0.67	1.76 ± 0.92	0.028	−0.39636
CRP, mg/dL	9.4 ± 8.4	13.6 ± 11.0	0.005	−0.43681
Procalcitonin, ng/mL	0.11 (0.08–0.21)	0.22 (0.11–0.70)	<0.001	0.06087
Lactate dehydrogenase, U/L	309.9 ± 131.3	368.1 ± 159.1	0.003	−0.39636
D-dimer, ng/mL	670 (430–1120)	980 (650–1950)	<0.001	−0.30192
High sensitivity troponin T > 14 ng/L	142 (30.7)	42 (77.8)	<0.001	1.07117
NT-proBNP, pg/mL	157 (39–417)	2065 (416–6780)	<0.001	−0.68728
Baseline QTc duration, milliseconds	430.2 ± 23.5	505.4 ± 31.9	<0.001	−2.68874
Abnormal chest radiography	495 (88.7)	54 (91.5)	0.51	0.09446

Results are expressed as mean ± standard deviation, median and (interquartile range) or number and (percentage). COPD: chronic obstruction pulmonary disease, CKD: chronic kidney disease, CRP: C-reactive protein. CV: Cardiovascular. eGFR: estimated glomerular filtration rate.

**Table 2 jcm-09-02712-t002:** Comparison of baseline characteristics and clinical presentation between patients alive and dead during hospitalization.

	Alive (*n* = 549)	Dead (*n* = 74)	*p*-Value	Standardized Differences
Women	230 (41.9)	37 (50.0)	0.486	0.16319
Age, years	62.0 ± 16.6	81.2 ± 8.6	<0.001	−1.45609
Diabetes	90 (16.4)	28 (37.8)	<0.001	0.49705
Hypertension	234 (42.6)	56 (75.7)	<0.001	0.71397
Dyslipidemia	175 (31.9)	37 (50.0)	0.002	0.37500
CV Risk factors	345 (62.8)	70 (94.6)	<0.001	0.95029
Obesity	96 (21.8)	16 (26.7)	0.40	0.11332
Ischemic chronic disease	31 (5.6)	20 (27.0)	<0.001	0.60412
Atrial fibrillation or flutter	35 (6.4)	17 (23.0)	<0.001	0.48252
Heart failure	18 (3.3)	15 (20.3)	<0.001	0.54652
Moderate to severe valve heart disease	14 (2.6)	9 (12.2)	<0.001	0.37460
COPD	38 (6.9)	14 (18.9)	<0.001	0.36353
Cancer	61 (11.1)	17 (23.0)	0.004	0.31947
CKD	33 (6.0)	24 (32.4)	<0.001	0.71171
Peripheral vascular disease	17 (3.1)	10 (13.5)	<0.001	0.38439
Stroke	26 (4.7)	9 (12.2)	0.009	0.26943
Systolic blood pressure, mmHg	128.2 ± 18.6	130.8 ± 25.4	0.400	−0.11696
Diastolic blood pressure, mmHg	77.4 ± 13.2	72.8 ± 16.1	0.022	0.31195
Heart rate, bpm	91.2 ± 17.9	88.5 ± 21.3	0.24	0.13723
Respiratory rate, rpm	24.1 ± 6.9	28.7 ± 7.1	<0.001	−0.66193
Oxygen saturation, %	95.2 ± 4.3	93.2 ± 5.9	0.006	0.38792
Baseline FiO2, %	25.1 ± 15.0	37.6 ± 26.7	<0.001	−0.57585
PaO2/FiO2 < 300, n (%)	165 (40.4)	45 (71.4)	<0.001	0.65697
Hemoglobin, g/dL	13.6 ± 1.6	12.6 ± 2.1	0.006	0.57111
White blood cell count, per µL	6.86 ± 3.05	9.99 ± 6.26	<0.001	−0.63711
Lymphocytes, per µL	1.46 ± 2.33	1.24 ± 1.55	0.43	0.11148
Platelet count	210.8 ± 77.9	207.5 ± 107.3	0.80	0.03488
Creatinine, mg/dL	0.99 ± 0.49	1.72 ± 1.46	<0.001	−0.66936
eGFR, mL/min/1.73 m^2^	84.0 ± 28.5	55.5 ± 31.9	<0.001	0.94242
Creatine phosphokinase, U/L	91 (57–167)	125 (49–290)	0.28	−0.33445
Serum lactate, mmoL/L	1.42 ± 0.66	1.79 ± 0.95	0.007	−0.45164
CRP, mg/dL	8.89 ± 7.78	16.41 ± 12.35	<0.001	−0.72780
Procalcitonin, ng/mL	0.11 (0.08–0.20)	0.26 (0.13–0.81)	<0.001	0.03618
Lactate dehydrogenase, U/L	1.42 ± 0.66	1.79 ± 0.95	<0.001	−0.66081
D-dimer, ng/mL	640 (420–1060)	1190 (780–3570)	<0.001	−0.47772
High sensitivity troponin T > 14 ng/L	135 (29.7)	49 (79.0)	<0.001	1.13899
NT-proBNP, pg/mL	144 (39–367)	1530 (550–4730)	<0.001	−0.67258
Baseline QTc duration, milliseconds	434.0 ± 27.8	464.3 ± 51.9	<0.001	−0.72743
QTc ≥ 480 ms	36 (6.6)	25 (33.8)	<0.001	0.72127
Abnormal chest radiography	481 (88.3)	68 (94.4)	0.115	0.22147

Results are expressed as mean ± standard deviation, median and (interquartile range) or number and (percentage). COPD: chronic obstruction pulmonary disease, CKD: chronic kidney disease, CRP: C-reactive protein. CV: Cardiovascular. eGFR: estimated glomerular filtration rate.

**Table 3 jcm-09-02712-t003:** Comparison of baseline characteristics, clinical presentation, treatment, and outcome between patients with and without QTc prolongation during hospitalization.

	No QTc Prolongation (*n* = 77)	QTc Prolongation (*n* = 168)	*p*-Value
Women	32 (41.6)	61 (36.3)	0.43
Age, years	62.1 ± 17.4	65.4 ± 15.0	0.14
Diabetes	10 (13)	40 (23.8)	0.051
Hypertension	32 (41.6)	96 (57.1)	0.023
Dyslipidemia	21 (27.3)	65 (38.7)	0.082
Obesity	13 (22.4)	42 (28.8)	0.36
Ischemic chronic disease	8 (10.4)	14 (8.3)	0.60
Atrial fibrillation or flutter	7 (9.1)	23 (13.7)	0.31
Heart failure	4 (5.2)	10 (6.0)	0.54
Moderate to severe valve heart disease	2 (2.6)	6 (3.6)	0.54
COPD	4 (5.2)	21 (12.5)	0.058
Cancer	13 (16.9)	29 (17.3)	0.94
CKD	5 (5.2)	14 (8.3)	0.28
Peripheral vascular disease	7 (9.1)	5 (3.0)	0.04
Stroke	2 (2.6)	13 (7.7)	0.097
Systolic blood pressure, mmHg	123.0 ± 18.8	128.4 ± 19.6	0.074
Diastolic blood pressure, mmHg	76.2 ± 14.7	75.2 ± 14.0	0.60
Heart rate, bpm	89.0 ± 15.0	91.2 ± 20.6	0.35
Respiratory rate, rpm	24.4 ± 6.4	26.2 ± 7.8	0.13
Oxygen saturation, %	95.9 ± 2.9	93.2 ± 6.3	<0.001
Baseline FiO2, %	24.9 ± 14.3	30.4 ± 22.3	0.022
PaO2/FiO2 < 300, n (%)	12 (21.4)	82 (58.2)	<0.001
Hemoglobin, g/dL	13.4 ± 1.5	13.6 ± 1.7	0.36
White blood cell count, per µL	6.3 ± 2.2	7.8 ± 3.6	<0.001
Lymphocytes, per µL	1.56 ± 2.60	1.37 ± 1.81	0.51
Platelet count	205.1 ± 72.6	202.6 ± 75.0	0.81
Creatinine, mg/dL	1.04 ± 0.51	1.15 ± 0.73	0.26
eGFR, mL/min/1.73 m^2^	81.1 ± 28.9	77.3 ± 29.7	0.34
Creatine phosphokinase, U/L	82 (50–156)	108 (63–181)	0.051
Serum lactate, mmoL/L	1.35 ± 0.59	1.55 ± 0.70	0.04
CRP, mg/dL	8.5 ± 7.7	11.6 ± 9.7	0.009
Procalcitonin, ng/mL	0.12 (0.07–0.17)	0.13 (0.08–0.25)	0.31
Lactate dehydrogenase, U/L	293.0 ± 92.5	348.8 ± 151.2	0.01
D-dimer, ng/mL	630 (410–940)	760 (470–1240)	0.027
High sensitivity troponin T > 14 ng/L	20 (33.9)	60 (41.4)	0.34
NT-proBNP, pg/mL	145 (42–303)	246 (76–756)	0.014
Baseline QTc duration, milliseconds	441.8 ± 38.5	435.38 ± 31.6	0.17
Baseline QTc ≥ 480 ms	8 (10.4)	20 (11.9)	0.73
Abnormal chest radiography	69 (90.8)	156 (92.9)	0.58
Tocilizumab	14 (18.2)	54 (32.1)	0.023
No hydroxychloroquine or azithromycin	6 (7.8)	1 (0.6)	0.015
Hydroxychloroquine alone	2 (2.6)	1 (1.8)	
Azithromycin alone	1 (1.3)	1 (0.6)	
Hydroxychloroquine + Azithromycin	68 (88.3)	163 (97.0)	
Oxygen support	46 (59.7)	91 (54.2)	<0.001
High Flow Nasal Cannula	2 (2.6)	5 (3.0)	
Non-invasive ventilation	1 (1.3)	16 (9.5)	
Intubation and invasive ventilation	1 (1.3)	37 (22.0)	
Longest QTc duration, milliseconds	440.4 ± 37.9	476.3 ± 48.9	<0.001
Length of hospitalization, days	10 (6–18)	17 (9–29.5)	<0.001
Dead	5 (6.5)	25 (14.9)	0.045

Results are expressed as mean ± standard deviation, median and (interquartile range) or number and (percentage). COPD: chronic obstruction pulmonary disease, CKD: chronic kidney disease, CRP: C-reactive protein. eGFR: estimated glomerular filtration rate.

**Table 4 jcm-09-02712-t004:** In-hospital treatment and outcome in patients with and without prolonged QTc on admission.

	QTc ≤ 480 (*n* = 562)	QTc > 480 (*n* = 61)	*p*-Value
Tocilizumab	94 (16.7)	11 (18.0)	0.80
No hydroxychloroquine or azithromycin	9 (1.6)	7 (11.5)	<0.001
Hydroxychloroquine	15 (2.7)	1 (1.6)	
Azithromycin	7 (1.2)	3 (4.9)	
Hydroxychloroquine + Azithromycin	531 (94.5)	50 (82.0)	
Oxygen support	337 (60.0)	40 (65.6)	0.12
High Flow Nasal Cannula	9 (1.6)	1 (1.6)	
Non-invasive ventilation	27 (4.8)	3 (4.9)	
Intubation and invasive ventilation	36 (6.4)	8 (13.1)	
QTc prolongation during hospitalization	148 (68.2)	20 (71.4)	0.73
Longest QTc duration, milliseconds	455.8 ± 40.5	536.5 ± 47.8	<0.001
Length of hospitalization, days	10 (5–19)	10 (3–23)	0.68
Death	49 (8.7)	25 (41.0)	<0.001

Results are expressed as mean ± standard deviation, median and (interquartile range) or number and (percentage).

**Table 5 jcm-09-02712-t005:** Comparison of in-hospital treatment and outcomes between patients alive and dead during hospitalization.

	Alive (*n* = 549)	Dead (*n* = 74)	*p*-Value
Tocilizumab	96 (17.5)	9 (12.2)	0.25
No hydroxychloroquine or azithromycin	9 (1.6)	7 (9.5)	<0.001
Hydroxychloroquine	12 (2.2)	4 (5.4)	
Azithromycin	6 (1.1)	4 (5.4)	
Hydroxychloroquine + Azithromycin	522 (95.1)	59 (79.7)	
Oxygen support	324 (59.0)	53 (71.6)	<0.001
High Flow Nasal Cannula	9 (1.6)	1 (1.4)	
Non-invasive ventilation	20 (3.6)	8 (10.8)	
Intubation and invasive ventilation	33 (6.0)	11 (14.9)	
QTc prolongation during hospitalization	143 (66.5)	25 (83.3)	0.045
Longest QTc duration, milliseconds	460.3 ± 45.3	499.0 ± 58.2	<0.001
Length of hospitalization, days	10 (5–20)	6 (3–16)	0.007

Results are expressed as mean ± standard deviation, median and (interquartile range) or number and (percentage).

**Table 6 jcm-09-02712-t006:** Hazard ratios (HR) of 30-day death for baseline QTc > 480 ms adjusted for potential confounders.

	Univariate HR (95%CI)	Adjusted HR (95%CI) *
Age (per every year)	1.01 (1.08–1.12), *p* < 0.001	1.08 (1.06–1.11), *p* < 0.001
Baseline QTc > 480 ms	4.87 (2.98–7.96), *p* < 0.001	2.68 (1.58–4.55), *p* < 0.001
Chronic kidney disease	6.07 (3.70–10.05), *p* < 0.001	2.62 (1.55–4.46), *p* < 0.001
Treatment with azithromycin and hydroxychloroquine	0.12 (0.05–0.26), *p* < 0.001	0.31 (0.13–0.72), *p* = 0.007
Ischemic chronic disease	3.60 (2.13–6.09), *p* < 0.001	-
Atrial fibrillation or flutter	3.04 (1.74–5.30), *p* < 0.001	-
Heart failure	5.43 (3.05–9.66), *p* < 0.001	-
Any cardiovascular risk factor	7.09 (2.58–19.45), *p* < 0.001	-

* Model adjusted for age, baseline QTc > 480 ms, chronic kidney disease, treatment with azithromycin and hydroxychloroquine, ischemic chronic disease, atrial fibrillation or flutter, heart failure, and the presence of any cardiovascular risk factor.

**Table 7 jcm-09-02712-t007:** Comparison of baseline characteristics, clinical presentation, treatment, and outcome between patients with and without ECG on presentation.

	Baseline ECG (*n* = 623)	Without Baseline ECG (*n* = 249)	*p*-Value
Women	267 (42.9)	119 (47.8)	0.19
Age, years	64.2 ± 17.0	57.3 ± 19.6	<0.001
Diabetes	118 (18.9)	55 (22.19)	0.29
Hypertension	290 (46.5)	93 (37.3)	0.013
Dyslipidemia	69 (27.7)	212 (34.0)	0.07
Any CV risk factor	415 (66.6)	150 (60.2)	0.08
Obesity	112 (22.4)	46 (21.8)	0.86
Ischemic chronic disease	51 (8.2)	9 (3.6)	0.016
Atrial fibrillation or flutter	52 (8.3)	18 (7.2)	0.58
Heart failure	33 (5.3)	10 (4.0)	0.43
Moderate to severe valve heart disease	23 (3.7)	6 (2.4)	0.34
COPD	52 (8.3)	14 (5.6)	0.17
Cancer	78 (12.5)	33 (13.3)	0.77
CKD	57 (9.1)	18 (7.2)	0.36
Peripheral vascular disease	27 (4.3)	7 (2.8)	0.29
Stroke	35 (5.6)	16 (6.4)	0.65
Systolic blood pressure, mmHg	128.5 ± 19.5	129.1 ± 19.4	0.68
Diastolic blood pressure, mmHg	76.9 ± 13.7	76.5 ± 13.5	0.72
Heart rate, bpm	90.9 ± 18.4	89.4 ± 16.4	0.27
Respiratory rate, rpm	24.6 ± 7.1	22.4 ± 8.2	0.011
Oxygen saturation, %	95.1 ± 4.4	93.8 ± 6.1	0.13
Baseline FiO2, %	26.6 ± 17.2	24.9 ± 8.2	0.42
PaO2/FiO2 < 300, n (%)	210 (44.6)	41 (41.4)	0.56
Hemoglobin, g/dL	13.5 ± 1.7	13.4 ± 1.8	0.34
White blood cell count, per µL	7.23 ± 3.7	6.31 ± 2.8	0.001
Lymphocytes, per µL	1.44 ± 2.26	1.48 ± 1.18	0.71
Platelet count	210.4 ± 81.8	220.5 ± 86.5	0.13
Creatinine, mg/dL	1.08 ± 0.72	0.95 ± 0.46	0.002
eGFR, mL/min/1.73 m^2^	80.7 ± 30.3	88.3 ± 32.4	0.003
Creatine phosphokinase, U/L	93 (57–175)	78 (50–151)	0.079
Serum lactate, mmoL/L	1.3 (1.03–1.67)	1.25 (0.93–1.6)	0.16
CRP, mg/dL	7.3 (3.1–14.1)	4.0 (1.5–9.3)	<0.001
Procalcitonin, ng/mL	0.12 (0.08–0.24)	0.09 (0.05–0.20)	<0.001
Lactate dehydrogenase, U/L	315.9 ± 135.1	295.6 ± 160.6	0.15
D-dimer, ng/mL	690 (440–1190)	610 (370–910)	0.004
High sensitivity troponin T > 14 ng/L	184 (35.7)	41 (30.4)	0.25
NT-proBNP, pg/mL	180 (48–540)	97 (25–346)	0.013
Abnormal chest radiography	549 (89.0)	161 (72.2)	<0.001
Tocilizumab	105 (16.9)	14 (5.6)	<0.001
No hydroxychloroquine or azithromycin	16 (2.6)	60 (24.1)	<0.001
Hydroxychloroquine	16 (2.6)	11 (4.4)	
Azithromycin	10 (1.6)	11 (4.4)	
Hydroxychloroquine + Azithromycin	581 (93.3)	167 (67.1)	
Oxygen support	377 (60.5)	78 (31.3)	<0.001
High Flow Nasal Cannula	10 (1.6)	3 (1.2)	
Non-invasive ventilation	28 (4.5)	4 (1.6)	
Intubation and invasive ventilation	44 (7.1)	11 (4.4)	
Length of hospitalization, days	10 (5–19)	8 (1–17)	<0.001
Death	74 (11.9)	27 (10.8)	0.66

Results are expressed as mean ± standard deviation, median and (interquartile range) or number and (percentage). CV: Cardiovascular. COPD: chronic obstruction pulmonary disease, CKD: chronic kidney disease, CRP: C-reactive protein. eGFR: estimated glomerular filtration rate.

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
