# Peer review of "Prolonged QT Interval in SARS-CoV-2 Infection: Prevalence and Prognosis"

_jcm, 2020, doi:10.3390/jcm9092712_

Round 1
Reviewer 1 Report
Comments and suggestions:
This is a well-written paper with a dense but informative results section, describing the prognostic value of a prolonged baseline QTc (>480 ms) for 623 consecutive in-hospital patients with PCR-confirmed SARS-Cov2 infection. In particular, the data on comorbidities, laboratory markers and treatments are extensive and excellent. As ECGs are cheap and noninvasive, using baseline QT prolongation as a prognostic marker could be of significant value.
I have only one major issue with this study, which pertains to the methods chosen to assess the QT interval.
Firstly, the QT intervals were automatically calculated using the Philips PageWriter TC30 Cardiograph. While the consistent use of one system for all patient ECGs is excellent, there are numerous studies on the difficulty of QT interval assessment, and the inconsistencies derived from automatic measurements, especially for prolonged QT intervals. If a completely automated analysis is chosen, either a manual measurement of a subset, or at least a visual confirmation of the ECGs (to check for consistency) should be considered (or objective data on the performance of the automated system presented). I would also like to see this issue raised in the discussion (automated vs. manual assessment of QT interval), if automated QTc measurements are proposed as a prognostic tool for SARS-Cov2.
Secondly, the authors have chosen the >480 ms cut-off for prolonged QTc, derived from the 2015 ESC Guidelines for the Management of Patients With Ventricular Arrhythmias and the Prevention of Sudden Cardiac Death. However, there are well documented sex-dependent differences in normal QT interval. Goldenberg (Goldenberg I, Moss AJ and Zareba W. QT interval: how to measure it and what is "normal". J Cardiovasc Electrophysiol. 2006;17:333-6.) suggested age- and sex- specific cut-offs for prolonged QTc (>450 ms for men and >470 ms for women). Based on this, I would have liked to see some gender specific data, as well as a discussion on the sex-dependence of QTc, and the risk of under-detecting male QTc prolongation when using the >480 ms cut-off.
Finally, if QTc prolongation within 48 hours of PCR-confirmed SARS-Cov2 infection indeed is secondary to the infection/inflammation, it would be interesting to see whether QTc levels normalize after recovery?
Author Response
This is a well-written paper with a dense but informative results section, describing the prognostic value of a prolonged baseline QTc (>480 ms) for 623 consecutive in-hospital patients with PCR-confirmed SARS-Cov2 infection. In particular, the data on comorbidities, laboratory markers and treatments are extensive and excellent. As ECGs are cheap and noninvasive, using baseline QT prolongation as a prognostic marker could be of significant value.
Thank you very much for your kind words.
I have only one major issue with this study, which pertains to the methods chosen to assess the QT interval.
Firstly, the QT intervals were automatically calculated using the Philips PageWriter TC30 Cardiograph. While the consistent use of one system for all patient ECGs is excellent, there are numerous studies on the difficulty of QT interval assessment, and the inconsistencies derived from automatic measurements, especially for prolonged QT intervals. If a completely automated analysis is chosen, either a manual measurement of a subset, or at least a visual confirmation of the ECGs (to check for consistency) should be considered (or objective data on the performance of the automated system presented). I would also like to see this issue raised in the discussion (automated vs. manual assessment of QT interval), if automated QTc measurements are proposed as a prognostic tool for SARS-Cov2.
We fully agree with the reviewer on this point. QT interval is difficult to assess, even for cardiologist. Thus, we thought that the best way to ensure that measures were consistent was to use the automated analysis. Moreover, several studies have shown that automated QTc measurements revealed accurate in comparison with manual QTc measurements (Bun et al, Clin Pharmacol Ther. 2020 Jun 26;10.1002/cpt.1968; Hooks et al. Heart Rhythm. 2020 Jun 28, Burke et al. J Electrocardiol. May-Jun 2014;47(3):288-93). We have added this limitation in the text (page 12, line 336).
Secondly, the authors have chosen the >480 ms cut-off for prolonged QTc, derived from the 2015 ESC Guidelines for the Management of Patients With Ventricular Arrhythmias and the Prevention of Sudden Cardiac Death. However, there are well documented sex-dependent differences in normal QT interval. Goldenberg (Goldenberg I, Moss AJ and Zareba W. QT interval: how to measure it and what is "normal". J Cardiovasc Electrophysiol. 2006;17:333-6.) suggested age- and sex- specific cut-offs for prolonged QTc (>450 ms for men and >470 ms for women). Based on this, I would have liked to see some gender specific data, as well as a discussion on the sex-dependence of QTc, and the risk of under-detecting male QTc prolongation when using the >480 ms cut-off.
We agree with the reviewer that a QTc cut-off of 480 ms might not be optimal for men.
When analyzed by sex, patients with QTc ≥ 480 ms had higher mortality. In women, mortality was 56.7% (17/30) in those with QTc ≥ 480 ms compared with 8.4% (20/237) in the non-prolonged QTc interval, p<0.001. Similar results were seen in men: mortality was 25.8% (8/31) in the prolonged QTc interval group vs. 8.9% (29/325) in the non-prolonged QTc interval group, p=0.003. This cut-off was independently associated with death in women (univariable HR 8.53 (95% CI: 4.45-16.36), p<0.001, multivariable HR 4.04 (1.98-8.27), p<0.001), whereas there was a strong tendency in the same direction in men (univariate HR 2.27 (95% CI: 0.99-5.23), p=0.053). It is worth mentioning that the number of events was very low in men (only 8 patients in the QTc ≥ 480 ms died) and that might explain the lack of statistically significance in men. Although the use a different cut-off according to sex could be useful, using several cut-off points depending on sex might not feasible in clinical practice when different types of healthcare professionals at several levels of complexity are involved.
We have included this information on the Results and Discussion section (page 7, line 212; and page 11, line 297).
Finally, if QTc prolongation within 48 hours of PCR-confirmed SARS-Cov2 infection indeed is secondary to the infection/inflammation, it would be interesting to see whether QTc levels normalize after recovery?
We agree with the reviewer. We are conducting a post-discharge study in which patients will have an ECG at 6 months after discharge to see if QTc values normalize.
Reviewer 2 Report
This is an interesting study that posits that QT interval could be elevated at baseline and associated with infection prior to treatment. A few recommendations
- the introduction regarding the link between QT and infection is fine. However, I believe there needs to be a further discussion between QT and other clinical characteristics that put people at risk of COVID-19. Based on the baseline univariate stats, there are many comorbid conditions associated with elevated QTc. Thus, it may be that the infection is not causing elevation, but the QT interval is just associated indirectly. Please expand upon this for balance.
- Please don't bury results in the supplemental table. I'd be interested in seeing these added to the main paper since there are no page limits with JCM.
- Table 3 is unclear and possible not adding up with the treatments. Instead indicate clearly among those that received HCQ alone, AZM alone, or HCQ+AZM. Same for Table 4.
- I am wholly uncomfortable including treatment variables in the regression analysis. There is a survival bias associated with treatment (or an immortal time bias) wherein you most survive long enough to be treated. With deaths having a shorter length of stay, this is one indictor of this bias. I would recommend using other variables besides these treatment variables.
- Analyses are multi-"variable" not "variate"
- Again I would like to see a baseline characteristics table for those that DID NOT receive an ECG to understand if there is additional selection bias added into the main text. Could also run a survival analysis to see the effect of "receiving an ECG" on survival to understand this.
- I am not comfortable supporting stepwise regression techniques as these ignore confounding and relationships between variables. I would hope that (although you are limited to probably ~7 variables based on # of events) you would select those that have very strong links between QT and death based on Table 1 and a literature search (e.g. heart failure, CKD, etc.).
- Would rather see standardized differences for Tables 1/2 to understand the scale of the difference rather than meaningless p-values. This may also help aid in selecting the strongest variables in the above point.
Author Response
This is an interesting study that posits that QT interval could be elevated at baseline and associated with infection prior to treatment.
A few recommendations.
The introduction regarding the link between QT and infection is fine. However, I believe there needs to be a further discussion between QT and other clinical characteristics that put people at risk of COVID-19. Based on the baseline univariate stats, there are many comorbid conditions associated with elevated QTc. Thus, it may be that the infection is not causing elevation, but the QT interval is just associated indirectly. Please expand upon this for balance.
We have expanded this part of the Discussion as per the reviewer suggestion (page 11, line 287).
Please don't bury results in the supplemental table. I'd be interested in seeing these added to the main paper since there are no page limits with JCM.
We have added the Supplemental table 1 and 2 in the main text, but summarized in one Table (Table 3).
Table 3 is unclear and possible not adding up with the treatments. Instead indicate clearly among those that received HCQ alone, AZM alone, or HCQ+AZM. Same for Table 4.
The review is right. We have modified the treatment on all the tables as no-treatment, HCQ alone, AZM alone and HCQ+AM.
I am wholly uncomfortable including treatment variables in the regression analysis. There is a survival bias associated with treatment (or an immortal time bias) wherein you most survive long enough to be treated. With deaths having a shorter length of stay, this is one indictor of this bias. I would recommend using other variables besides these treatment variables.
We agree with the reviewer that adding a treatment is not the best option. However, since in this case the treatment with HCQ and AZM are known to prolong QT and might predispose to ventricular arrhythmias, we think that its inclusion is warranted. However, we did two things to try to minimize the bias. First, we carried out a model with the same variables except that did not include the treatment and the results were similar (HR 2.78 (95% CI 1.66-4.66), p<0.001). We also did a sensitivity analysis excluding patients who died during the first 48 hours of admission (18 patients, 24.3% of all patients who died) and the results were also similar (HR: 2.073 (95% CI: 1.073-4.005), p=0.03).
We have added this information on the Methods and Results sections (page 4, line 124).
Analyses are multi-"variable" not "variate".
Thank you for the correction. We have changed it in the manuscript.
Again I would like to see a baseline characteristics table for those that DID NOT receive an ECG to understand if there is additional selection bias added into the main text. Could also run a survival analysis to see the effect of "receiving an ECG" on survival to understand this.
We have added this analysis on the main text (page 2, line 74 and page 9 line 246). There was no difference in the death rate between patients with and without a baseline ECG. There was a mistake in the number shown in the Figure 1 that has been corrected.
I am not comfortable supporting stepwise regression techniques as these ignore confounding and relationships between variables. I would hope that (although you are limited to probably ~7 variables based on # of events) you would select those that have very strong links between QT and death based on Table 1 and a literature search (e.g. heart failure, CKD, etc.).
We agree with the reviewer and this is how we did the analysis. We have describe it better in the Methods section (page 4, line 124).
Would rather see standardized differences for Tables 1/2 to understand the scale of the difference rather than meaningless p-values. This may also help aid in selecting the strongest variables in the above point.
As suggested, standardized differences were calculated, and a difference >0.10 was considered clinically significant. This information has been included in Table 1 and 2 and in the Methods section (page 4, line 124). All variables included in the multivariable analysis had a difference >0.1. Female sex had stddiff >0.1 but p>0.2 when compared between QTc cut-off and survival. Univariate Cox showed a HR 0.24 (95% CI 0.49-1.08), p=0.11. Therefore, we did not include it in the multivariable model.
Round 2
Reviewer 2 Report
Much improved. Best of luck.